# Learning Guidance Rewards with Trajectory-space Smoothing

**Tanmay Gangwani**
Dept. of Computer Science
UIUC
gangwan2@illinois.edu

**Yuan Zhou**
Dept. of ISE
UIUC
yuanz@illinois.edu

**Jian Peng**
Dept. of Computer Science
UIUC
jianpeng@illinois.edu

## Abstract

Long-term temporal credit assignment is an important challenge in deep reinforcement learning (RL). It refers to the ability of the agent to attribute actions to consequences that may occur after a long time interval. Existing policy-gradient and Q-learning algorithms typically rely on dense environmental rewards that provide rich short-term supervision and help with credit assignment. However, they struggle to solve tasks with delays between an action and the corresponding rewarding feedback. To make credit assignment easier, recent works have proposed algorithms to learn dense *guidance* rewards that could be used in place of the sparse or delayed environmental rewards. This paper is in the same vein – starting with a surrogate RL objective that involves smoothing in the trajectory-space, we arrive at a new algorithm for learning guidance rewards. We show that the guidance rewards have an intuitive interpretation, and can be obtained without training any additional neural networks. Due to the ease of integration, we use the guidance rewards in a few popular algorithms (Q-learning, Actor-Critic, Distributional-RL) and present results in single-agent and multi-agent tasks that elucidate the benefit of our approach when the environmental rewards are sparse or delayed [1].

## 1 Introduction

Deep Reinforcement Learning (RL) agents are tasked with maximization of long-term environmental rewards. Prevalent algorithms for deep RL typically need to estimate the expected future rewards after taking an action in a particular state – Actor-critic and Q-learning involve computing the Q-value, while policy-gradient methods tend to be more stable when using the advantage function. The value estimation is performed using temporal difference (TD) or Monte-Carlo (MC) learning. Although deep RL algorithms can achieve remarkable results on a wide variety of tasks, their performance crucially depends on the meticulously designed reward function, which provides a dense per-timestep learning signal and facilitates value estimation. In real-world sequential decision-making problems, however, the rewards are often sparse or delayed. Examples include, to name a few, industrial process control [9], molecular design [17], and resource allocation [19]. Delayed rewards introduce high bias in TD-learning and high variance in MC-learning [1], leading to poor value estimates. This impedes long-term *temporal credit assignment* [15, 32], which refers to the ability of the agent to attribute actions to consequences that may occur after a long time interval. As a motivating example in a simulated domain, Figure 1 shows the performance with Soft-Actor-Critic (SAC) [7], a popular off-policy RL method, on two MuJoCo locomotion tasks from the Gym suite. For *delay*=$k$, the agent receives no reward for $(k-1)$ timesteps and is then provided the accumulated rewards at the $k^{\text{th}}$ timestep. Increasing the delay leads to progressively worse performance.

Another class of policy search algorithms, which are particularly handy when rewards are delayed,

is black-box stochastic-optimization; examples include Evolution Strategies [22] and Deep-Neuroevolution [30]. They are invariant to delayed rewards since the trajectories are not decomposed into individual timesteps for learning; rather zeroth-order finite-difference or gradient-free methods are used to learn policies based only on the trajectory returns (or aggregated rewards). However, one downside is that discarding the temporal structure of the RL problem leads to inferior sample-efficiency when compared with the standard RL algorithms. Our goal in this paper is to design an approach that easily integrates into the

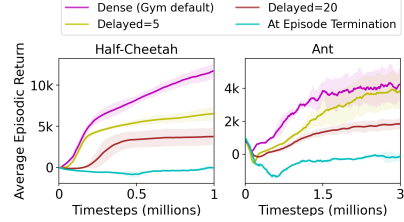

Figure 1: Effect of delayed rewards

existing RL algorithms, thus enjoying the sample-efficiency benefits, while being invariant to delayed rewards. To achieve this, we introduce a surrogate RL objective that involves smoothing in the trajectory-space and arrive at a new algorithm for learning *guidance* rewards. The guidance rewards are curated using only the trajectory returns and are easily inferred for each state-action tuple. The dense supervision from the guidance rewards makes value estimation and credit assignment easier, substantially accelerating learning when the original environmental rewards are sparse or delayed.

We provide an intuitive understanding for the guidance rewards in terms of *uniform* credit assignment – they characterize a uniform redistribution of the trajectory return to each constituent state-action pair. A favorable property of our approach is that no additional neural networks need to be trained to obtain the guidance rewards, unlike recent works that also examine RL with delayed rewards (*c.f.* Section 5). For quantitative evaluation, we combine the guidance rewards with a variety of RL algorithms and environments. These include single-agent tasks: Q-learning [33] in a discrete grid-world and SAC on continuous control locomotion tasks; and multi-agent tasks: TD3 [4] and Distributional-RL [2] in multi-particle cooperative environments.

## 2 Background and Notations

Our RL environment is modeled as an infinite-horizon, discrete-time Markov Decision Process (MDP). The MDP is characterized by the tuple $(\mathcal{S}, \mathcal{A}, r, p, \gamma)$, where $\mathcal{S}$ and $\mathcal{A}$ are the state- and action-space, respectively, and $\gamma \in [0, 1)$ is the discount factor. Given an action $a_t$, the next state is sampled from the transition dynamics distribution, $s_{t+1} \sim p(s_{t+1}|s_t, a_t)$, and the agent receives a reward $r(s_t, a_t)$ determined by the reward function $r : \mathcal{S} \times \mathcal{A} \to \mathbb{R}$. A stochastic policy $\pi(a_t|s_t)$ defines the state-conditioned distribution over actions. $\tau$ denotes a trajectory $\{s_0, a_0, s_1, a_1, \dots\}$ and $R(\tau)$ is the sum of discounted rewards over the trajectory, $R(\tau) = \sum_{t=0}^{\infty} \gamma^t r(s_t, a_t)$. The RL objective is to learn $\pi$ that maximizes the expected $R(\tau)$, $\eta(\pi) = \mathbb{E}_{p,\pi}[R(\tau)]$.

**Actor-critic Algorithms.** These methods use a critic for value function estimation and an actor that is updated based on the information provided by the critic. The critic is trained with TD-learning in a policy-evaluation step; then the actor is updated with an approximate gradient in the direction of policy improvement. Under certain conditions, their repeated application converges to an optimal policy [31]. We briefly outline two model-free off-policy actor-critic RL algorithms that work extremely well on high-dimensional tasks and are used in this paper – TD3 and SAC. TD3 is a deterministic policy gradient algorithm (DPG) [25]. It uses a deterministic policy $\mu_\theta$ that is updated with the policy gradient: $\nabla_\theta \mathbb{E}_{s \sim \rho^\beta}[Q^\mu(s, \mu_\theta(s))]$, where $\rho^\beta$ is the state distribution of a behavioral policy $\beta$, and $Q^\mu$ is the state-action value trained with the Bellman error. TD3 alleviates the Q-function overestimation bias in DPG by using Clipped Double Q-learning when calculating the Bellman target. Differently, SAC optimizes for the maximum entropy RL objective, $\mathbb{E}_\pi[\sum_t \gamma^t(r(s_t, a_t) + \alpha \mathcal{H}(\pi(\cdot|s_t)))]$, where $\mathcal{H}$ and $\alpha$ are the policy entropy and the temperature, respectively. SAC alternates between soft policy evaluation, which estimates the soft Q-function using a modified Bellman operator, and soft policy improvement, which updates the actor by minimizing the Kullback-Leibler divergence between the policy distribution and exponential form of the soft Q-function. The loss functions for the critic ($Q_\phi$), the actor ($\pi_\theta$) and the temperature ($\alpha$) are:

$$J_Q(\phi; r) = \mathbb{E}_{\substack{(s,a,s') \sim \mathcal{D} \\ a' \sim \pi_\theta(\cdot|s')}} \Big[\frac{1}{2}\big(Q_\phi(s, a) - (r(s, a) + \gamma(Q_{\bar\phi}(s', a') - \alpha \log \pi_\theta(a'|s')))\big)^2\Big] \quad (1)$$

$$J_\pi(\theta) = \mathbb{E}_{\substack{s \sim \mathcal{D} \\ a \sim \pi_\theta(\cdot|s)}} \big[\alpha \log(\pi_\theta(a|s)) - Q_\phi(s, a)\big]; \quad J(\alpha) = \mathbb{E}_{\substack{s \sim \mathcal{D} \\ a \sim \pi_\theta(\cdot|s)}} \big[-\alpha \log \pi_\theta(a|s) - \alpha \bar{\mathcal{H}}\big] \quad (2)$$

where $\mathcal{D}$ is the replay buffer, $Q_{\bar\phi}$ is the target critic network, and $\bar{\mathcal{H}}$ is the expected target entropy.

# 3 Method

This section begins with the definition of our modified RL objective that involves smoothing in the trajectory-space, following which we make design choices that result in guidance rewards. Given a policy $\pi_\theta$, the standard RL objective is: $\arg\max_\theta \mathbb{E}_{\tau \sim \pi(\theta)}[R(\tau)]$. As motivated before, with delayed environmental rewards, directly optimizing this objective hinders learning due to difficulty in temporal credit assignment caused by value estimation errors. Objective function smoothing has long been studied in the stochastic optimization literature. In the context of RL, Salimans et al. [22] proposed Evolution Strategies (ES) for policy search. ES creates a smoothed version of the standard RL objective using *parameter-level* smoothing (usually Gaussian blurring):

$$\eta^{\text{ES}}(\pi_\theta) = \mathbb{E}_{\epsilon \sim \mathcal{N}(0,I)}\mathbb{E}_{\tau \sim \pi(\theta + \sigma \cdot \epsilon)}[R(\tau)]$$

where $\sigma$ controls the level of smoothing. Although ES is invariant to delayed rewards, eschewing the temporal structure of the RL problem often results in low sample efficiency. Following the broad principle of using a smoothed objective to obtain effective gradient signals, we consider explicit smoothing in the *trajectory-space*, rather than the parameter-space. We define our maximization objective as:

$$\eta_{\text{smooth}}(\pi_\theta) = \mathbb{E}_{\hat{\tau} \sim \pi(\theta)}\big[\mathbb{E}_{\tau \sim M_{\hat{\tau}}}[R(\tau)]\big] \tag{3}$$

where $M_{\hat{\tau}}(\tau)$ is the smoothing distribution over trajectories $\tau$ that is parameterized by the reference trajectory $\hat{\tau}$. When $M$ is a delta distribution, *i.e.*, $M_{\hat{\tau}}(\tau) = \delta(\tau = \hat{\tau})$, the original RL objective is recovered. We wish to design a smoothing distribution $M$ that helps with credit assignment. Let $\beta(a|s)$ denote a behavioral policy and the trajectory distribution induced by $\beta$ in the MDP be $p_\beta(\tau) = p(s_0)\prod_{t=0}^{\infty} p(s_{t+1}|s_t, a_t)\beta(a_t|s_t)$. Further, we introduce $p_\beta(\tau; s, a)$ as the distribution over the $\beta$-induced trajectories which *include* the state-action pair $(s, a)$:

$$p_\beta(\tau; s, a) \stackrel{\text{def}}{=} \frac{p_\beta(\tau)\mathbb{1}[(s,a) \in \tau]}{\int_\tau p_\beta(\tau)\mathbb{1}[(s,a) \in \tau]\,\mathrm{d}\tau}$$

where $\mathbb{1}$ is the indicator function. For consistency, we require that the normalization constant be positive $\forall (s, a)$. Let $\{\hat{s}_t, \hat{a}_t\}$ be the reference state-action pairs in the reference trajectory $\hat{\tau}$. We propose the following infinite mixture model for the smoothing distribution $M_{\hat{\tau}}(\tau)$:

$$M_{\hat{\tau},\beta}(\tau) = (1 - \gamma)\sum_{t=0}^{\infty} \gamma^t p_\beta(\tau; \hat{s}_t, \hat{a}_t)$$

Given a reference trajectory $\hat{\tau}$, this distribution samples trajectories from the behavioral policy $\beta$ that *intersect* or overlap with the reference trajectory, with intersections at later timesteps discounted exponentially with the factor $\gamma$. Inserting this in Equation 3, rearranging and ignoring constants, the smoothed objective to maximize becomes:

$$\eta_{\text{smooth}}(\pi_\theta) = \mathbb{E}_{\hat{\tau} \sim \pi(\theta)}\Big[\sum_{t=0}^{\infty} \gamma^t \underbrace{\int_\tau p_\beta(\tau; \hat{s}_t, \hat{a}_t)R(\tau)\,\mathrm{d}\tau}_{r_{\text{g}}(\hat{s}_t, \hat{a}_t)}\Big] \tag{4}$$

This is equivalent to the standard RL objective, albeit with a different reward function than the environmental reward. We define this as the guidance reward function, $r_{\text{g}}(s, a; \beta) = \mathbb{E}_{\tau \sim p_\beta(\tau; s, a)}[R(\tau)]$. The guidance reward apportioned to each state-action pair is the expected value (under $p_\beta$) of the returns of the trajectories which *include* that state-action pair. Useful features of $r_{\text{g}}$ are that it provides a dense reward signal, and is invariant to delays in the environmental rewards since it depends on the trajectory return. Thus, it potentially promotes better value estimation and credit assignment.

**Interpretation as uniform credit assignment.** Temporal Credit assignment deals with the question: *"given a final outcome (e.g. trajectory returns), how relevant was each state-action pair in that trajectory towards achieving the return?"*. Prior work has proposed learning estimators that explicitly model the relevance of an action to future returns, or using contribution analysis methods to redistribute rewards to the individual timesteps (*c.f.* Section 5). Our method could be viewed as performing a simple redistribution – it *uniformly* distributes the trajectory return among the state-action pairs in that trajectory. [2] This non-committal or maximum entropy credit assignment is natural to consider in

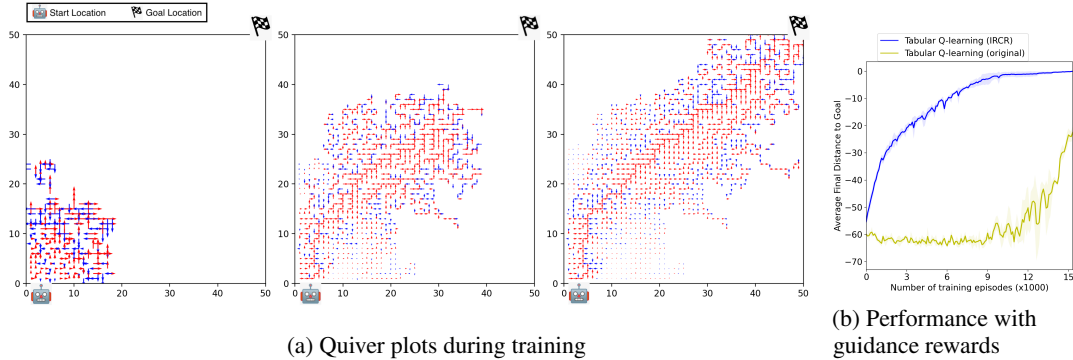

(a) Quiver plots during training
(b) Performance with guidance rewards

Figure 2: We consider a $50 \times 50$ grid-world with the start and goal locations marked in the image. The rewards are episodic – a non-zero reward is only provided at the end of the episode (horizon=150 steps) and is equal to the negative of the distance of the final position to the goal. We run for 15k episodes. The three quiver plots (left to right) are taken after 100 episodes, 2k episodes and 15k episodes, respectively. In each quiver plot, an arrow in a state represents the guidance reward: the direction denotes the action with *maximum* guidance reward, *i.e.,* $\arg\max_a r_{\mathrm{g}}(s,a)$, and the length denotes its magnitude in $[0,1]$. A state with no arrow mean that the guidance reward is 0 for all actions in that state. For ease of exposition, we have colored all arrows pointing up/right with red and all arrows pointing down/left with blue. We note that over time, reasonable guidance rewards emerge along the diagonal path from the start location to the goal. Although the guidance rewards in the top-left and bottom-right regions of the grid are imprecise, they are not critical for learning the optimal policy to achieve the task. Figure (right) compares tabular Q-learning with environmental rewards and our guidance rewards. *Quiver plots best viewed when digitally zoomed.*

the absence of any prior structure or information. The guidance reward for each state-action pair is then obtained as the expected value (under $p_\beta$) of the uniform credit it receives from the different trajectory returns. For clarity of exposition, Appendix A.3 demonstrates the guidance rewards using some elementary MDPs and $p_\beta$.

### 3.1 Integrating guidance rewards into existing RL algorithms

Without access to the true MDP reward function, it is infeasible to solve for the guidance rewards exactly. Hence, we resort to a Monte-Carlo (MC) estimation for the expectation, $r_{\mathrm{g}}(s,a;\beta) = \mathbb{E}_{\tau \sim p_\beta(\tau; s, a)}[R(\tau)]$. Let $\mathbf{\Gamma}$ denote a fixed set of trajectories generated in the MDP using $\beta$. The MC estimate can be written as: $r_{\mathrm{g}}(s,a) = (1/N(s,a)) \sum_{\tau \in \mathbf{\Gamma}} \left[R(\tau)\mathbb{1}[(s,a) \in \tau]\right]$, where $N(s,a)$ is the number of trajectories in $\mathbf{\Gamma}$ which include the tuple $(s,a)$. Please see Appendix A.1 for details on achieving the MC estimate from the smoothed RL objective (Equation 4). It is possible to deploy an exploratory behavioral policy $\beta$ to obtain the set $\mathbf{\Gamma}$ in a pre-training phase. Following that, the *stationary* guidance rewards computed from $\mathbf{\Gamma}$ could be used to learn a new policy with any of the standard RL methods. One issue with this sequential approach is that it is challenging to design $\beta$ such that it achieves adequate state-action-space coverage in high dimensions. Perhaps more importantly, it is typically unnecessary to have good estimates for the guidance rewards for the *entire* state-action-space. For instance, in a grid-world, if the goal location is always to the right of the starting position of the agent, it is acceptable to have imprecise guidance reward in the left half of the grid, as long as the agent is discouraged from venturing to the left. Therefore, we propose an iterative approach where the experience gathered thus far by the agent is used to build the guidance rewards, *i.e.,* $r_{\mathrm{g}}(s,a)$ is the expected value of the uniform credit received by $(s,a)$ from the trajectories already rolled out in the MDP. Simultaneously, a policy $\pi$ (or Q-function) is learned using these *non-stationary* rewards. With this procedure, $\beta$ could be thought of as being implicitly defined as a mixture of current and past policies $\pi$. The scale of the credit apportioned to a state-action pair from a trajectory depends on the scale of its return value. For the guidance rewards to be effective, it is sufficient that the *relative* values of the rewards be properly aligned to solve the task. Therefore, when assigning credits, we normalize the trajectory returns to $[0,1]$ using min-max normalization. We refer to our approach for producing guidance rewards as **Iterative Relative Credit Refinement (IRCR)**. The guidance rewards are adapted over time as the average credit assigned to each state-action pair is refined by the information (return value) from newly sampled trajectories.

---

**Algorithm 1:** Tabular Q-learning with IRCR

---

1  Initialize Q(s,a) $\leftarrow$ 0
2  $R_{\max} \leftarrow -\infty$; $R_{\min} \leftarrow \infty$      ▷ Maximum/Minimum return thus far
3  $\mathcal{B}(s, a) \leftarrow \emptyset \ \forall \ (s, a)$  ▷ Buffer that stores for each $(s, a)$, a list of returns of trajectories that include that $(s, a)$

4  **Function** GetGuidanceReward$(s, a)$:
     /* Get normalized returns; return 0 if $\mathcal{B}(s, a) = \emptyset$ */
5    **return** $\mathbb{E}_{R_i \sim \mathcal{B}(s,a)} \left[ \frac{R_i - R_{\min}}{R_{\max} - R_{\min}} \right]$

6  **for** *each episode* **do**
7    $R_e \leftarrow 0$      ▷ Accumulates rewards for current episode
8    $\tau_e \leftarrow \emptyset$      ▷ Stores state-action pairs for current episode

9    **for** *each step in* $\{1, \ldots, T\}$ **do**
10      Choose $\mathbf{a}$ from $\mathbf{s}$ using policy derived from Q ($\epsilon$-greedy)
11      Take action $\mathbf{a}$ and observe $\mathbf{r}, \mathbf{s}'$      ▷ Sample transition from the environment
12      $\tau_e \leftarrow \tau_e \cup \{(\mathbf{s}, \mathbf{a})\}$;  $R_e \leftarrow R_e + \mathbf{r}$
13      $\mathbf{r}_g(\mathbf{s}, \mathbf{a}) \leftarrow$ GetGuidanceReward$(\mathbf{s}, \mathbf{a})$
14      Q$(\mathbf{s}, \mathbf{a}) \leftarrow$ Q$(\mathbf{s}, \mathbf{a}) + \alpha\,[\mathbf{r}_g(\mathbf{s}, \mathbf{a}) + \gamma \max_{\mathbf{a}'}$ Q$(\mathbf{s}', \mathbf{a}')$ - Q$(\mathbf{s}, \mathbf{a})]$
15    **end**

16    **for** *each (s,a) in* $\tau_e$ **do**
17      $\mathcal{B}(s, a) \leftarrow \mathcal{B}(s, a) \cup \{R_e\}$      ▷ Update $\mathcal{B}$ for $(s, a)$ along the collected trajectory
18    **end**
19    $R_{\max} \leftarrow \max(R_{\max}, R_e)$;  $R_{\min} \leftarrow \min(R_{\min}, R_e)$      ▷ Update $R_{\max}, R_{\min}$
20  **end**

---

Any standard RL algorithm could be modified by replacing the environmental rewards with the guidance rewards. In Algorithm 1, we outline this for tabular Q-learning with small state and action spaces. The notable change from the standard Q-learning is the use of $r_g$ in Line 14, instead of the environmental reward. To compute $r_g$, we maintain a buffer $\mathcal{B}(s, a)$ for each state-action pair that stores the returns of the past trajectories which include that state-action pair (Line 17). The guidance rewards evolve over time since the average credit allotted to a state-action pair changes as more experience is gathered in the MDP. To illustrate this, we run Algorithm 1 in a $50 \times 50$ grid-world with episodic environmental rewards. Figure 2a provides some insights on the structure of the guidance rewards assigned to the different regions of the grid as training progresses; the description of the episodic rewards and the arrows in the quiver plots is provided in the figure caption. In Figure 2b, we show the performance gains compared with tabular Q-learning using environmental rewards. Please see Appendix A.2 for hyperparameters and other details.

**Scaling to high-dimensional continuous spaces.** Actor-critic algorithms that scale to more complex environments (*e.g.* TD3, SAC) maintain an experience replay buffer [12] that stores $\{s, a, s', r\}$ tuples. These algorithms can be readily tailored to use guidance rewards. In Algorithm 2, we summarize SAC with IRCR. The environmental reward in the experience replay tuple is replaced with the return of the trajectory which produced that tuple (Line 13). When computing the soft Bellman error for learning the soft Q-function, the guidance reward is calculated by normalizing this return value (Lines 18-19). Mathematically, this is equivalent to the MC estimation of the guidance reward using a single trajectory, rather than the expected credit from a trajectory distribution. This is not an issue in practice if the soft Q-function, which is learned with these guidance rewards, is parameterized by a deep neural network that tends to generalize well in the vicinity of the input data. Indeed, as our experiments will show, Algorithm 2 achieves reliable performance in high-dimensional tasks. Other actor-critic algorithms could be modified analogously to incorporate the guidance rewards.

**Convergence.** Some comments are in order concerning the convergence of our iterative approach. We provide a qualitative analysis by drawing an analogy with the Cross Entropy (CE) method [14, 21]. For policy search, CE uses a multivariate Gaussian distribution to represent a population of policies. In each iteration, individuals $\pi_k$ are drawn from this distribution, their fitness, $\mathbb{E}_{\tau \sim \pi_k}[R(\tau)]$, is evaluated, and a fixed number of fittest individuals determine the new mean and variance of the population. This fitness-based selection ensures steady policy improvement. In IRCR, the trajectories

**Algorithm 2:** Soft Actor-Critic with IRCR

1 Initialize $\phi, \bar{\phi}, \theta$      ▷ Policy and critic parameters
2 $R_{\max} \leftarrow -\infty; R_{\min} \leftarrow \infty$      ▷ Maximum/Minimum return thus far
3 $\mathcal{D} \leftarrow \emptyset$      ▷ Empty replay buffer
4 **for** *each episode* **do**
5    $R_e \leftarrow 0$      ▷ Accumulates rewards for current episode
6    $\mathcal{D}_e \leftarrow \emptyset$      ▷ Stores transitions for current episode
7    **for** *each step in* $\{1, \ldots, T\}$ **do**
8      $\mathbf{a} \sim \pi_\theta(\mathbf{a}|\mathbf{s})$
9      Take action $\mathbf{a}$ and observe $\mathbf{r}, \mathbf{s}'$      ▷ Sample transition from the environment
10      $\mathcal{D}_e \leftarrow \mathcal{D}_e \cup \{(\mathbf{s}, \mathbf{a}, \mathbf{s}')\}; \ \ R_e \leftarrow R_e + \mathbf{r}$
11    **end**
12    **for** *each* $(s, a, s') \in \mathcal{D}_e$ **do**
13      $\mathcal{D} \leftarrow \mathcal{D} \cup \{(s, a, s', R_e)\}$      ▷ Append each transition with $R_e$ and add to replay buffer
14    **end**
15    $R_{\max} \leftarrow \max(R_{\max}, R_e); \ \ R_{\min} \leftarrow \min(R_{\min}, R_e)$      ▷ Update $R_{\max}, R_{\min}$
16    **for** *each gradient step* **do**
17      $\{s^{(k)}, a^{(k)}, s'^{(k)}, R^{(k)}\}_{k \in \mathbb{N}^+} \sim \mathcal{D}$      ▷ Sample batch
18      $\mathbf{r_g^{(k)}} \leftarrow \frac{R^{(k)} - R_{\min}}{R_{\max} - R_{\min}}$      ▷ Get guidance reward by normalizing return
19      $\phi \leftarrow \phi - \lambda \nabla_\phi J_Q(\phi; \mathbf{r_g})$      ▷ Update Q-function using guidance rewards, (Eq. 1)
20      $\theta \leftarrow \theta - \lambda \nabla_\theta J_\pi(\theta); \ \alpha \leftarrow \alpha - \lambda \nabla_\alpha J(\alpha)$      ▷ Update policy and temperature, (Eq. 2)
21    **end**
22 **end**

generated by a mixture of the current and past policies ($\pi_{0:i}$) are used to compute the guidance rewards ($r_\mathrm{g}$); $\pi_{i+1}$ is then obtained by a policy optimization step with these rewards. Since $r_\mathrm{g}$ is positively correlated with the environmental returns $R(\tau)$, maximizing for a discounted sum of $r_\mathrm{g}$ tends to seek out a policy that attains higher $R(\tau)$ compared to $\pi_{0:i}$, on average. Consequently, this optimization step facilitates policy improvement in the same spirit as the CE method. The next section provides empirical evidence that the policy behavior improves over iterations of IRCR. We consider the theoretical study of convergence as an important future work.

## 4 Experiments

This section evaluates our approach on various single-agent and multi-agent RL tasks to quantify the benefits of using the guidance rewards in place of the environmental rewards, when the latter are sparse or delayed.

### 4.1 Single-agent environments and baselines

We benchmark high-dimensional, continuous-control locomotion tasks based on the MuJoCo physics simulator, provided in OpenAI Gym [3]. We compare SAC (IRCR) outlined in Algorithm 2 with the following baselines:

- SAC with environmental rewards. It uses the same hyperparameters as SAC (IRCR). Please see Appendix A.2 for details.

- Generative Adversarial Self-imitation Learning (*GASIL*), which represents the method proposed in Guo et al. [6]; Gangwani et al. [5]. A buffer stores the top-$k$ trajectories according to the return. A discriminator network, which is a binary classifier that distinguishes the buffer data from data generated by the current policy, acts as a source of the guidance rewards.

- *Reward Regression*, which typifies the approaches presented in Arjona-Medina et al. [1]; Liu et al. [13]. They formulate a regression task that predicts the return given the entire trajectory. A network trained with this regression loss helps to decompose the trajectory return back to the constituent state-action pairs, and provides the guidance rewards. We include the results from Liu et al. [13] using the Transformer architecture (data obtained from authors).

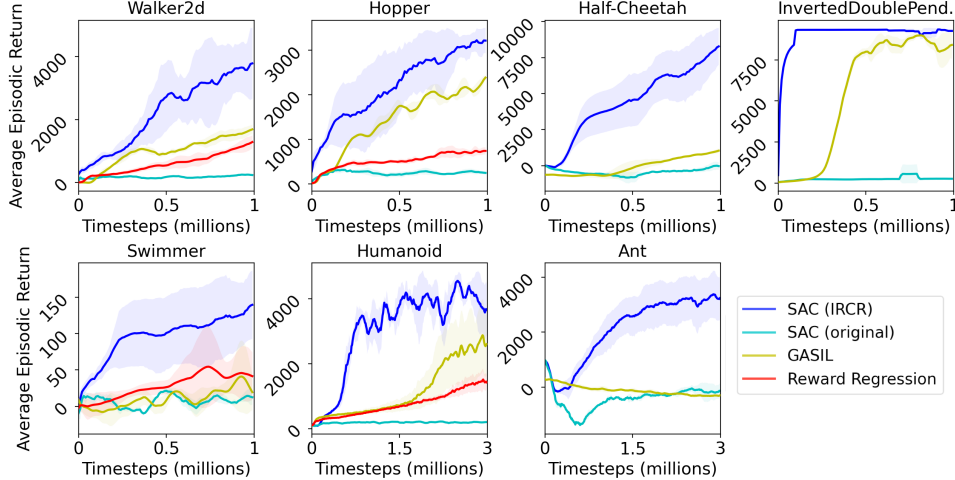

Figure 3: Learning curves for the MuJoCo locomotion tasks with **episodic rewards**. The mean and standard deviation over 5 random seeds are plotted. Reward-regression baseline has some missing data[3].

We modify the reward function in Gym to design tasks with episodic rewards – the agent collects a zero reward for all timesteps *except* the last one, at which the accumulated original reward is given. With reference to Figure 1 in the Introduction, this expresses the maximum possible delay in rewards. Figure 3 plots the learning curves for all the algorithms with episodic rewards. We observe SAC (IRCR) to be the most sample-efficient across all tasks. SAC (original) is unable to learn any useful behavior since the value estimation errors due to delayed environmental rewards impede temporal credit assignment. GASIL and Reward-regression [3] are invariant to delayed rewards, but the learning is much more sluggish than our approach, possibly due to the instability in training the additional neural network for reward function estimation (discriminator for GASIL and Transformer for Reward-regression). In contrast, IRCR does not require any auxiliary networks and simply defines the guidance rewards as the expected value of the credit apportioned to $(s, a)$ from past trajectories.

### 4.2 Multi-agent environments and baselines

We evaluate IRCR in a sparse-reward cooperative multi-agent RL (MARL) environment. This setting involves multiple agents that execute actions that jointly influence the environment; the agents receive local observations and a shared (sparse) reward. We adopt the *Rover Domain* from Rahmattalabi et al. [20] in which agents navigate in a two-dimensional world with continuous states and actions. There are $N$ rovers (agents) and $K$ Points-of-interests (POIs). A global reward is achieved when any POI is *harvested*. For harvesting a POI, a certain minimum number of rovers—determined by a *Coupling* parameter—need to be simultaneously within a small observation radius around that POI; higher couplings require greater coordination. Figure 4b illustrates a scenario with coupling=2. Each rover has sensors to detect other rovers and POI around it using a mechanism similar to a LIDAR. A rover within the observation radius of an un-harvested POI receives a small local reward.

Our baseline algorithm is MA-TD3, a multi-agent extension of TD3 (Section 2) with the critic network shared among all agents. This is compared with two methods that employ guidance rewards – MA-TD3 (IRCR), which replaces the environmental rewards with the guidance rewards (similar to Algorithm 2), but uses the same underlying update rules; and MA-C51 (IRCR), which replaces the Clipped Double Q-learning in MA-TD3 with the C51 distributional-RL algorithm [2]. Our C51 variant includes other minor alternations detailed in Appendix A.4. We experiment with different values for $N, K$, and the coupling factor (Appendix A.2). Figure 4a plots, for coupling factors 1 to 4, the percentage of the POIs that are harvested at the end of an episode vs. the number of training timesteps. We note that MA-TD3 (IRCR) is more sample-efficient and achieves higher scores compared to MA-TD3 with environmental rewards, that are sparse since the agent collects a zero reward if it is outside the observation radius of every POI. Finally, the good performance of MA-C51 (IRCR) suggests that guidance rewards can be effectively used with distributional-RL as well.

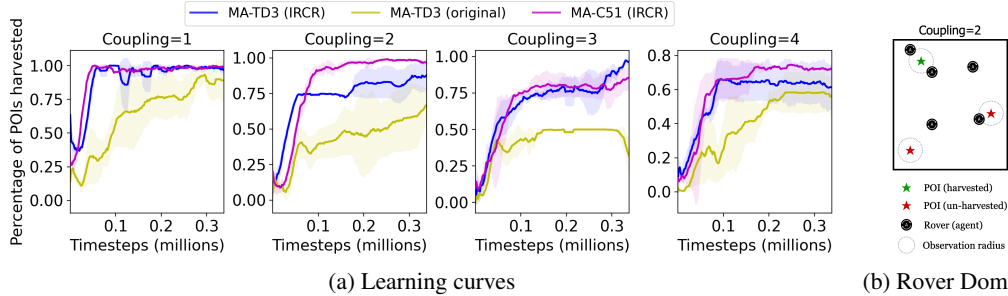

Figure 4: (a) Learning curves for the Rover domain with different coupling factors. The mean and standard deviation over 5 random seeds are plotted.; (b) Rover Domain (illustrated with coupling=2).

## 5    Related work

**Reward shaping.** Designing reward functions that modify or substitute the original rewards is a popular method for improving the learning efficiency of RL agents. Ng et al. [16] developed potential-based shaping functions that guarantee the preservation of the optimal policy. Providing bonus rewards inspired by intrinsic motivation ideas such as curiosity [23, 24] has been shown to aid exploration. The optimal reward problem (ORP) [26, 27] introduces the idea that the original reward (that captures the *designer's* intent) could be decoupled from the rewards used by the agent in the RL algorithm (guidance rewards), even though the agent is finally evaluated on the designer's intent. Using the guidance rewards in place of the original rewards can substantially accelerate learning, especially if the agent is bounded by computational or knowledge constraints [28, 29]. IRCR could be interpreted as an instance of ORP. It provides dense guidance rewards that improve value estimation when the original environmental rewards are sparse or delayed, thus enabling faster learning.

**Credit Assignment.** A variety of methods have been proposed to improve temporal credit assignment. Hindsight Credit Assignment [8] learns a model that quantifies the relevance of an action to a future outcome, such as the total return following that action. An interesting feature of the model is that the Q-function estimate for *all* the actions could be improved using the returns sampled from a certain starting action. Temporal Value Transport [10] uses an attention-based memory module that links past events (at time $t'$) to the present time $t$. The state-value at $t$ is then "transported" to $t'$, to be used as an additional bootstrap (or fictitious reward) in the TD error at $t'$. This helps in efficiently propagating credit backward in time. Neural Episodic Control [18] and Episodic Backward Update [11] enable faster propagation of sparse or delayed rewards from the entire episode through all the transitions of the episode. In RUDDER [1], an LSTM network is trained to predict the trajectory return at every time-step. The guidance reward is then obtained as the difference of consecutive predictions. Liu et al. [13] use a Transformer with masked multi-head self-attention as the learned reward function. It is trained by regressing on the trajectory-return and helps to decompose the return back to each time-step in the trajectory. Adversarial self-imitation approaches [5, 6] use a min-max objective to train a discriminator and a policy iteratively. The discriminator is learned with a binary classification loss and provides guidance rewards for policy optimization. In contrast with these, our computation of the guidance rewards does not require training auxiliary networks and could be viewed as a simple uniform return decomposition.

## 6    Conclusion

In this paper, we introduce a surrogate RL objective with smoothing in the trajectory-space. We show that our choice of the smoothing distribution makes this objective equivalent to standard RL, albeit with the guidance reward function instead of the environmental reward. The guidance rewards are easily measurable for any state-action pair as the expected return of the past trajectories which include that pair. The dense supervision afforded by them makes value estimation and temporal credit assignment easier. Our method is invariant to delayed rewards, does not require training auxiliary networks, and integrates well with existing RL algorithms. Experimental results across a variety of RL algorithms with single- and multi-agent tasks highlight the contribution of the guidance rewards in improving the sample-efficiency, especially when the environmental rewards are sparse or delayed.

## Broader Impact

In this paper, we propose techniques to improve the sample-efficiency of Reinforcement Learning (RL) algorithms when the environmental rewards are sparse or delayed. Many real-world decision-making problems of interest are of this nature – the rewarding (or penalizing) feedback is usually available only after a long sequence of interaction with the environment. Some prominent examples include a.) *Chemical Synthesis*, where the product yield and the functional measurements mostly happen at the last step of the entire process; b.) *Industrial Control Processes*, which involve sequential calibration of many control variables to achieve the desired output, for instance, the final metal purity metric in a metal-refining furnace operation; and c.) *RL in healthcare* for uncovering promising treatment regimes, where there could be delayed interaction between treatments and human bodies. Techniques that make RL algorithms robust in the face of delayed rewards are therefore poised to have a hugely positive societal influence across a range of sectors. We believe our work is a step in the direction of developing such techniques.

## Acknowledgments and Disclosure of Funding

This work is supported by the National Science Foundation under grants OAC-1835669 and CCF-2006526. Yuan Zhou is supported in part by a Ye Grant and a JPMorgan Chase AI Research Faculty Research Award.

## Footnotes

[1]Code for this paper is available at https://github.com/tgangwani/GuidanceRewards

[2]In Equation 4, we excluded the constant $(1 - \gamma)$ from the smoothing distribution $M_{\hat{\tau}}(\tau)$ to reduce clutter. Since $1/(1 - \gamma)$ is the effective horizon, $(1 - \gamma)R(\tau)$ represents a uniform redistribution of the trajectory return to each constituent state-action pair.

[3]We were unable to obtain the data for *InvertedDoublePendulum*, *Ant* and *Half-Cheetah* for Reward-regression from the authors of Liu et al. [13] since they do not evaluate on these three tasks.

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
