[Supplementary Material]

## A  **Appendix:** Learning Guidance Rewards with Trajectory-space Smoothing

### A.1  **Monte-Carlo Estimate of the Guidance Rewards**

$$\eta_{\text{smooth}}(\pi_\theta) = \mathbb{E}_{\hat{\tau} \sim \pi(\theta)} \Big[ \sum_{t=0}^{\infty} \gamma^t \mathbb{E}_{\tau \sim p_\beta(\tau; s, a)}[R(\tau)] \Big]; \quad p_\beta(\tau; s, a) \overset{\text{def}}{=} \frac{p_\beta(\tau) \mathbb{1}[(s, a) \in \tau]}{\int_\tau p_\beta(\tau) \mathbb{1}[(s, a) \in \tau] \, d\tau}$$

Let $\rho_\pi(s, a)$ denote the unnormalized discounted state-action visitation distribution for $\pi$. Then:

$$\eta_{\text{smooth}}(\pi_\theta) = \mathbb{E}_{(s,a) \sim \rho_\pi} \mathbb{E}_{\tau \sim p_\beta(\tau; s, a)}[R(\tau)]$$

Plugging in the definition of $p_\beta(\tau; s, a)$ and using linearity of expectations:

$$\eta_{\text{smooth}}(\pi_\theta) = \mathbb{E}_{\tau \sim p_\beta(\tau)} \mathbb{E}_{(s,a) \sim \rho_\pi} \left[ \frac{R(\tau) \mathbb{1}[(s, a) \in \tau]}{\int_\tau p_\beta(\tau) \mathbb{1}[(s, a) \in \tau] \, d\tau} \right]$$

Let $\Gamma$ denote a set of $N$ trajectories generated in the MDP using $\beta$, and $N(s, a)$ be the count of the trajectories which include the tuple $(s, a)$. The Monte-Carlo estimate of $\eta_{\text{smooth}}(\pi_\theta)$ is:

$$\hat{\eta}_{\text{smooth}}(\pi_\theta) = \frac{1}{N} \sum_{\tau \in \Gamma} \mathbb{E}_{(s,a) \sim \rho_\pi} \left[ \frac{R(\tau) \mathbb{1}[(s, a) \in \tau]}{N(s, a)/N} \right]$$

$$= \mathbb{E}_{(s,a) \sim \rho_\pi} \underbrace{\sum_{\tau \in \Gamma} \left[ \frac{R(\tau) \mathbb{1}[(s, a) \in \tau]}{N(s, a)} \right]}_{r_{\text{g}}(s,a)}$$

where $r_{\text{g}}(s, a)$ is same as the Monte-Carlo estimate of the guidance rewards defined in Section 3.1. We further define $r_{\text{g}}(s, a) = 0$ if $N(s, a) = 0$.

### A.2  **Implementation Details and Hyperparameters**

**Tabular Q-learning**

Table 1 provides the hyperparameters for the Tabular Q-learning experiment in a $50 \times 50$ grid-world with episodic environmental rewards (Figure 2). Both IRCR and the baseline share the same hyperparameters.

| Hyperparameter | Value |
|---|---|
| learning rate | 0.3 at start, linearly annealed |
| exploration | $\epsilon$-greedy ($\epsilon = 0.5$ at start, linearly annealed) |
| discount ($\gamma$) | 0.9 |
| episode horizon | 150 steps |
| # training episodes | 15000 |

Table 1: Hyper-parameters for Tabular Q-learning

**SAC**

Table 2 provides the hyperparameters for SAC (Algorithm 2) which is used for the experiments in Figure 3. For the experience replay, the usual FIFO buffer is augmented with a small Min-Heap buffer which stores a few high return past episodes (details in Table). This is used in all the baselines as well. Both SAC (IRCR) and SAC (original) share the same hyperparameters.

| Hyperparameter | Value |
| --- | --- |
| # hidden layers (all networks) | 2 |
| # hidden units per layer | 256 |
| # samples per minibatch | 256 |
| nonlinearity | ReLU |
| optimizer | Adam [3] |
| discount $(\gamma)$ | 0.99 |
| entropy target | $-|\mathcal{A}|$ |
| target smoothing coefficient | 0.001 |
| learning rate | $1 \times 10^{-4}$ for policy and temperature, $3 \times 10^{-4}$ for critic |
| replay buffer | $3 \times 10^5$ transitions (FIFO) + 10 episodes (Min Heap) |

Table 2: Hyper-parameters for SAC

**MA-TD3**

Since our multi-agent task is cooperative and involves maximization of a scalar team reward, we use a single critic network that is shared amongst the agents. Furthermore, as the agents are homogeneous (same observation- and action-space), we design a permutation-invariant critic:

$$Q(\mathbf{s}, \mathbf{a}) = g(\text{mean}(\{f(o_1, a_1), \ldots, f(o_k, a_k)\}))$$

where $(o_i, a_i)$ are the local observation and action of agent $i$, $(\mathbf{s}, \mathbf{a})$ denote joint observations and actions, and $g, f$ are neural networks. This permutation invariant set representation was proposed by Zaheer et al. [6]. Table 3 provides the hyperparameters for MA-TD3 which is used for the experiments in Figure 4a. Both MA-TD3 (IRCR) and MA-TD3 (original) share the same hyperparameters.

| Hyperparameter | Value |
| --- | --- |
| policy architecture | 2 hidden layers, 128 hidden units |
| critic architecture | shared, permutation-invariant (described above) |
| # samples per minibatch | 256 |
| nonlinearity | Tanh |
| optimizer | Adam [3] |
| discount $(\gamma)$ | 0.99 |
| target smoothing coefficient | 0.005 |
| episode horizon | 100 steps |
| learning rate | $1 \times 10^{-4}$ for policy, $3 \times 10^{-4}$ for critic |
| replay buffer | $5 \times 10^4$ transitions (FIFO) + 100 episodes (Min Heap) |
| exploration strategy | $\mathcal{N}(0, 0.1)$ + Adaptive Parameter Noise [4] |

Table 3: Hyper-parameters for MA-TD3

The following are the specifics for the environments used in Figure 4a:

- **Coupling=1.** # POIs = 3, # Agents = 3
- **Coupling=2.** # POIs = 4, # Agents = 4
- **Coupling=3.** # POIs = 4, # Agents = 6
- **Coupling=4.** # POIs = 4, # Agents = 8

## A.3 Illustration of Guidance Rewards with Simple MDP and $p_\beta$

Consider the MDP in Figure 5 with the states $\{s_1, s_2, s_3, s_4\}$, $s_1$ is the start state, $\{s_3, s_4\}$ are the terminal states. $\{a_1, a_2\}$ are the possible actions from $s_1$; $\{a_3, a_4\}$ are the possible actions from $s_2$. There are 4 possible trajectories. Let the return associated with each trajectory be the following:

Figure 5: MDP with 4 states and 4 actions

- $\tau_1 : \{s_1 a_1 s_2 a_3\}; R(\tau_1) = 1$
- $\tau_2 : \{s_1 a_1 s_2 a_4\}; R(\tau_2) = 3$
- $\tau_3 : \{s_1 a_2 s_2 a_3\}; R(\tau_3) = 1$
- $\tau_4 : \{s_1 a_2 s_2 a_4\}; R(\tau_4) = 1$

The guidance reward is given by:

$$r_g(s, a; \beta) = \mathbb{E}_{\tau \sim p_\beta(\tau; s, a)}[R(\tau)] \qquad p_\beta(\tau; s, a) \overset{\text{def}}{=} \frac{p_\beta(\tau) \mathbb{1}[(s, a) \in \tau]}{\int_\tau p_\beta(\tau) \mathbb{1}[(s, a) \in \tau]\, d\tau}$$

We compute the guidance rewards for the above MDP for two different $p_\beta$ distributions - uniform and exponential.

**With Uniform $p_\beta$**

If $p_\beta$ is uniform, $p_\beta(\tau_1) = p_\beta(\tau_2) = p_\beta(\tau_3) = p_\beta(\tau_4) = 0.25$. From this, we obtain:

- $p_\beta(\tau; s_1, a_1) = \frac{1}{2}\delta(\tau = \tau_1) + \frac{1}{2}\delta(\tau = \tau_2); \quad r_g(s_1, a_1; \beta) = 2$
- $p_\beta(\tau; s_1, a_2) = \frac{1}{2}\delta(\tau = \tau_3) + \frac{1}{2}\delta(\tau = \tau_4); \quad r_g(s_1, a_2; \beta) = 1$
- $p_\beta(\tau; s_2, a_3) = \frac{1}{2}\delta(\tau = \tau_1) + \frac{1}{2}\delta(\tau = \tau_3); \quad r_g(s_2, a_3; \beta) = 1$
- $p_\beta(\tau; s_2, a_4) = \frac{1}{2}\delta(\tau = \tau_2) + \frac{1}{2}\delta(\tau = \tau_4); \quad r_g(s_2, a_4; \beta) = 2$

**With Exponential $p_\beta$**

If $p_\beta$ is exponential, *i.e.* $p_\beta(\tau) \propto \exp(R(\tau))$, $p_\beta(\tau_1) = p_\beta(\tau_3) = p_\beta(\tau_4) = 0.1$; $p_\beta(\tau_2) = 0.7$ (rounded off to 1 decimal). From this, we obtain:

- $p_\beta(\tau; s_1, a_1) = \frac{1}{8}\delta(\tau = \tau_1) + \frac{7}{8}\delta(\tau = \tau_2); \quad r_g(s_1, a_1; \beta) = 2.75$
- $p_\beta(\tau; s_1, a_2) = \frac{1}{2}\delta(\tau = \tau_3) + \frac{1}{2}\delta(\tau = \tau_4); \quad r_g(s_1, a_2; \beta) = 1$
- $p_\beta(\tau; s_2, a_3) = \frac{1}{2}\delta(\tau = \tau_1) + \frac{1}{2}\delta(\tau = \tau_3); \quad r_g(s_2, a_3; \beta) = 1$
- $p_\beta(\tau; s_2, a_4) = \frac{7}{8}\delta(\tau = \tau_2) + \frac{1}{8}\delta(\tau = \tau_4); \quad r_g(s_2, a_4; \beta) = 2.75$

## A.4 Distributional-RL with Guidance Rewards

### Background

Distributional-RL models the full distribution of the returns, the expectation of which is the Q-function. We use the C51 algorithm introduced by Bellemare et al. [1] which represents the return distribution with learned probabilities on a fixed support; several other representation methods have also been proposed. Let $Z^\pi(s, a)$ be the random variable denoting the sum of discounted rewards along a trajectory starting with the state-action pair $(s, a)$. The value function is $Q^\pi(s, a) = \mathbb{E}Z^\pi(s, a)$. $Z^\pi(s, a)$ is obtained by the repeated application of the distributional Bellman operator $\mathcal{T}^\pi$ defined as:

$$\mathcal{T}^\pi Z(s, a) \overset{D}{=} R(s, a) + \gamma Z(s', a') \quad s' \sim p(\cdot|s, a), a' \sim \pi(\cdot|s')$$

C51 models the value distribution with a discrete distribution on a fixed support $\{z_i\}_{i=1}^N$, referred to as a set of *atoms*. The atom probabilities are given by a learned parametric model $f_\theta : \mathcal{S} \times \mathcal{A} \to \mathbb{R}^N$:

$$Z_\theta(s, a) = z_i \quad w.p. \quad p_\theta^i(s, a) = \text{softmax}(f_\theta(s, a))_i$$

**Atom Support and Guidance Rewards in Log-space**

In Bellemare et al. [1], the support of the atoms ranges from $V_{\min}$ to $V_{\max}$, which are environment-specific variables defining the limits of the returns possible in that environment. To make the support range environment-agnostic, we define it in the log-space: $w_i = \{1/N, 2/N, \ldots, 1\}$ and $z_i = \log w_i$. Thus, the Q-function is written as $Q_\theta(s, a) = \sum_i p_\theta^i(s, a) \log w_i$.

We further define guidance rewards modified with a log function, $r_{\text{Lg}}(s, a) = \log r_{\text{g}}(s, a)$. Recall that $r_{\text{g}} \in [0, 1]$ due to the min-max normalization; hence the application of $\log$ is proper (*expect at 0 where a small $\epsilon$ should be added*). Although this transformation changes the magnitude, the relative ordering of the guidance rewards is preserved due to the monotonicity of the $\log$. The parametric model $f_\theta$ is optimized with TD-learning. With the distributional Bellman equation, this is equivalent to a distribution matching problem. Given a training tuple $(s, a, r_{\text{Lg}}, s')$ from the replay buffer, the discrete target distribution is:

$$r_{\text{Lg}} + \gamma z_i \quad w.p. \quad p_\theta^i = \text{softmax}(f_\theta(s', a'))_i$$

Using the log-space atom support and definition of $r_{\text{Lg}}$, we can rewrite this as:

$$\log\left[r_{\text{g}}.w_i^{\gamma}\right] \quad w.p. \quad p_\theta^i = \text{softmax}(f_\theta(s', a'))_i$$

Similarly, the discrete source distribution is:

$$\log w_i \quad w.p. \quad p_\theta^i = \text{softmax}(f_\theta(s, a))_i$$

The source and the target distributions have a support interval ($-\infty$, 0]. In principle, any $f$-divergence metric on them could be minimized. An alternative is to induce a transformation before the divergence minimization. This is justified by the following theorem from Qiao & Minematsu [5]: *"The f-divergence between two distributions is invariant under differentiable and invertible transformation"*. With an exponential transformation, we get the following distributions that are now shaped to have a support interval (0,1]:

$$r_{\text{g}}.w_i^{\gamma} \quad w.p. \quad p_\theta^i = \text{softmax}(f_\theta(s', a'))_i$$
$$w_i \quad w.p. \quad p_\theta^i = \text{softmax}(f_\theta(s, a))_i$$

Following [1], we minimize the KL-divergence between these distributions using a *projection* step to account for the mismatch in the atom positions between the source and the target.

## A.5   Discussion Points

**Exploration vs. Credit-assignment.** Exploration and credit assignment are two distinct fundamental problems in RL. The former deals with the *discovery* of new useful information, the latter is about efficiently incorporating this information for learning a robust policy. In hard exploration problems, an agent typically obtains zero rewards in each episode unless an exploration impetus is given, whereas in our setting, a reward signal is readily provided to the agent at the end of *every* episode. The focus, therefore, is to train effectively from this delayed feedback and improve upon the credit assignment.

**Limitations of IRCR.** Since the reward that the agent optimizes for is different from the original task reward and is coupled to a behavioral policy $\beta$, for a given MDP, it might be possible to design an adversarial $\beta$ such that optimizing for the resultant guidance rewards leads to unintended behaviors (as per the task rewards). Although not visible in our empirical evaluation, a limitation of IRCR is that a careful adaptation of $\beta$ could be crucial in some domains to avoid this. For instance, if $\beta$ gets stuck in some region of the state-action-space, the learning agent may also get trapped in a local optimum due to *deceptive* guidance rewards. Combining IRCR with methods that explicitly incentivize exploration is a promising approach.

## A.6   Further Experiments

**Robotic manipulator arm environment.** Figure 6 shows the robotic arm that models a 7 degree-of-freedom Sawyer robot, inspired by Chen et al. [2]. The task is to insert a cylindrical peg (held in the end-effector attached to the arm) into a hole some distance away on the table. A non-zero reward is provided only at the end of every episode and is equal to exponential of negative $L_2$ distance between the final position of the peg and the hole. In Table 4, we compare the final performance of the SAC

Figure 6: MuJoCo model of a 7 DoF arm based on the Sawyer robot.

| | SAC (env. rewards) | SAC (IRCR) | Random Policy |
|---|---|---|---|
| | $104\pm4$ | $160\pm7$ | $90\pm11$ |

Table 4: SAC performance on the peg-insertion task with environmental and guidance rewards. Mean and standard deviation over 5 random seeds are reported.

| | SAC (env. rewards) $n$-step returns $(n=10)$ | SAC (env. rewards) $\lambda$-returns $(\lambda=0.9)$ | TD3 (env. rewards) | TD3 (IRCR) |
|---|---|---|---|---|
| *Hopper* | $626\pm281$ | $566\pm101$ | $235\pm37$ | $2981\pm114$ |
| *Half-Cheetah* | $-42\pm29$ | $-112\pm206$ | $-123\pm50$ | $6844\pm918$ |

Table 5: Performance (mean and standard deviation over 5 random seeds) of various algorithms.

algorithm with environmental and guidance rewards, and note that the latter is considerably better.

**Baselines for better reward propagation.** Table 5 includes the following baselines: SAC ($n$-step) uses $n$-step returns for the Bellman error update of the soft Q-function, and SAC ($\lambda$-returns) augments the Q-function training with TD($\lambda$), which interpolates nicely between 1-step TD and MC returns based on the value of $\lambda$. We tested with $n=\{3,10,20\}$ and $\lambda=\{0.5,0.9,0.95,1.0\}$ but observe that these methods are unable to solve the MuJoCo locomotion tasks with episodic rewards. Our conjecture for their low scores with episodic rewards is that the reward delay exacerbates the variance is MC, and bias in TD, to such an extent that using them separately, or mixing them via interpolation, is not sufficient to alleviate the problems in value estimation for credit assignment. IRCR takes a different approach of learning guidance rewards for each time-step and integrates well with 1-step TD (due to the bias reduction afforded by the dense guidance rewards). Table 5 also contrasts TD3 (IRCR) with TD3 (environmental rewards) on the locomotion tasks.