[Reviews · NeurIPS 2020]

Review 1

Summary and Contributions: This paper proposes the idea of replacing the actual rewards the agent sees by a transformation of the return of the whole trajectory. Specifically, given a trajectory, instead of using the reward at time step t in the update rule, this paper proposes one should use a reward signal that is the byproduct of uniformly distributing the trajectory return among the state-action pairs in that trajectory. This is said to improve agents' performance in face of "delayed environmental rewards". This change can be easily incorporated in pretty much any RL method and it is shown to outperform other approaches in MuJoCo tasks as well as in some multiagent problems.

Strengths: The paper is well-written, clear, and it does tackle an interesting problem. The introduced technique is quite simple and the experiments do depict the efficacy of the proposed method when compared to other baselines.

Weaknesses: This paper seems to ignore two important approaches for faster credit assignment in reinforcement learning. As discussed by Ostrovski et al. (2017), "it is well known that better performance, both in terms of learning efficiency and approximation error, is attained by multi-step methods (Sutton, 1996; Tsitsiklis & van Roy, 1997). These methods interpolate between one-step methods (Q-Learning) and the Monte-Carlo update." A simple instantiation of this method that is used throughout is the mixed monte-carlo return (see Ostrovski et al., 2017 for a careful discussion). Another technique equally relevant is the n-step returns so common in deep reinforcement learning. Hessel et al. (2018) recently discussed it and showed its benefits. This reference is just one of the several that use n-step returns. All in all, I don't understand why these two methods are not discussed and they are not used as baselines. My score is mostly tied to this limitation. Specifically, some of the questions this paper doesn't answer are: 1. How is IRCR different from Mixed Monte-Carlo returns and n-step returns? 2. What are its benefits? 3. How does it compare to these other approaches empirically? References: Georg Ostrovski, Marc G. Bellemare, Aäron van den Oord, Rémi Munos: Count-Based Exploration with Neural Density Models. ICML 2017: 2721-2730 Matteo Hessel, Joseph Modayil, Hado van Hasselt, Tom Schaul, Georg Ostrovski, Will Dabney, Dan Horgan, Bilal Piot, Mohammad Gheshlaghi Azar, David Silver: Rainbow: Combining Improvements in Deep Reinforcement Learning. AAAI 2018: 3215-3222

Correctness: Overall the paper seems correct to me. I have concerns about how the paper is positioning itself in the literature (discussed above) and how it intertwines the problem of delayed rewards and exploration (discussed below), but overall the paper seems correct to me.

Clarity: The way the paper discusses the problem of delayed rewards is a little bit confusing to me. My main issue is related to the distinction between delayed rewards and exploration. Are these problems the same? Exploration was never mentioned in the paper, but is rewarding the agent at the end of the episode a problem that highlights the exploration issues faced by our current agents? Clearly, "dense" reward functions could be seen as implementing some notion of reward shaping, which is discussed in Section 5, but I wonder what's the role of exploration in all that. Are the issues highlighted here due to the agent not being able to properly assign credit or are they due to the fact that without the "guiding" rewards (such as distance in MuJoCo) the agent rarely stumbles upon the goal? This seems an important distinction to make. Maybe a concrete question is: in the tasks used in this problem, does the agent observe the same reward at the end regardless of its actions? Or even more explicitly, given a fixed batch of data, would the agent learn faster using IRCR instead of a different technique such as Q-learning?

Relation to Prior Work: I already discussed this issue in the weaknesses section. This is in fact my main worry about this paper.

Reproducibility: Yes

Additional Feedback: Overall the paper is well-written and the experiments seem to be correct. The idea is simple and potentially promising. However, the concerns I have in terms of distinguishing the credit assignment problem from the exploration problem, and in terms of distinguishing IRCR from n-step returns and mixed-monte carlo returns, prevent me from giving the paper a higher score. I'll happily increase my score if the authors can clearly argue about the benefits/differences of IRCR over n-step returns and mixed monte-carlo returns, and if they are able to clarify the role of exploration in the problem studied. ----- Review after rebuttal ----- The authors' rebuttal addressed most of my concerns, specifically w.r.t. the new results about n-step returns and mixed monte-carlo returns. I recommend the authors to be clearer about credit assignment and exploration in the paper (as they were in the rebuttal). Also, aside from the experimental results, it would be interesting to see a discussion in the paper about the main conceptual differences between what was introduced and n-step returns / MMC. I'm now recommending the acceptance of this paper.


Review 2

Summary and Contributions: 1. The paper looks at the challenge of long term credit assignment in RL. The paper proposes a method to learn dense guidance rewards that help in value learning and credit assignment. 2. They basically redistribute the trajectory return uniformly to the different s,a pairs in the trajectory, thereby providing dense rewards. 3. The authors show significant improvement in performance in Mujoco locomotion tasks.

Strengths: 1. A very important practically relevant problem 2. The authors provide a simple solution and show considerably improved performance to not using their proposed method and also two other baseline ways to provide guidance rewards

Weaknesses: 1. The paper lacks discussion on the limitations of the proposed method. 2. In the current version of the proposed method, I think it is possible to attribute past rewards to future states. This is when there are indeed a few rewards other than the terminal rewards from the environment. I think it is possible to avoid that by just looking at only the future rewards while calculating the trajectory return in the calculation of guidance reward for that state,action pair 3. Given the change in agents objective, it becomes important to study how the solution of the changed objective can vary from the original objective (provided by the environment reward function). One can imagine the possibility of weird unintended / suboptimal policies / loops being learned because of the change in the reward function. 4. From the current evaluation, it is not clear to me where the proposed methods stands w.r.t to other past methods in the literature that try to handle long term credit assignment. (Some of which mentioned in the Related Works section in the paper, such as reward shaping, learning intrinsic rewards or RUDDER etc)

Correctness: 1. In cases where are there are multiple equally valid ways to solve the problem, and if we learn the guidance rewards for states before hand, it feels like the agent might learn unintended solutions. For example if there are two valid solutions through states a-> b -> c and a-> d -> e to solve the problem. Is it possible that this method could make the agent learn to do a -> b -> e? Which will not be rewarding by the environment, but the guidance rewards give equal return to that of a -> d -> e?

Clarity: Yes, but could be improved a bit.

Relation to Prior Work: 1. I am wondering why the authors have not compared their work to some of the relevant work that they have mentioned in their Related Works sections. They seem like approproriate baselines to compare with. I am wondering why the particular two baselines presented were selected compared to the others presented in the Related Works section.

Reproducibility: Yes

Additional Feedback: 1. If there are indeed dense rewards, not sure if the proposed method would maintain/help/reduce performance. Can you comment on it please. After author feedback: Thanks for your answering several of my concerns and questions. I am increasing my score based on that.


Review 3

Summary and Contributions: The authors consider a MDP setting in which the return is only observed at the end of the trajectory. To overcome the difficult credit assignment problem, the authors propose a method IRCR that retroactively redistributes the resulting return across the entire episode. Specifically, for every transition in a trajectory, the reward is replaced with the return obtained at the end of the episode. The authors demonstrate empirically that this simple approach outperforms prior methods that train neural networks to generate rewards across a variety of mujoco tasks.

Strengths: The main strength of this paper is the simplicity and strong results of this method. I think a number of reinforcement learning practitioners would find these results intriguing and the method potentially useful. The paper is fairly written, and the exact idea is relatively novel.

Weaknesses: While the authors provide some qualitative analysis (Section 3.1) about the convergence of the method, the paper would be substantially strengthened by having a more rigorous analysis that studies what this objective optimizes. A technical comment is that in the “delayed reward” setting, the problem is partially observed since the reward is now a function of the time. So, it is unsurprising that a method like SAC, which is developed for MDPs, performs poorly. I understand that addressing this limitation of SAC is the purpose of this paper, but it would help to separate the partial observability from the reward delay. For example, the authors could augment the state space with the cumulative sum of discounted rewards, at which point this problem is no longer partially observed. In that case, if SAC still performs poorly, we would know that SAC performs poorly only due to the partial observability, and not due to the delayed rewards. The empirical results would be strengthened by including the MA-C51 (original) baseline as well as including the results of Reward Regression on all the MuJoCo tasks.

Correctness: See Weaknesses for comments on empirical results.

Clarity: The paper is generally easy to follow. While the paper tries to present an interpretation of their method as a mixture of objectives, it is a bit unclear why this mathematical interpretation is useful, especially given the lack of analysis. Nit-pick: “For consistency, we require that the normalization constant be positive ∀(s, a).” isn’t it always positive? “it is acceptable to have imprecise guidance reward in the left half of the grid, as long as the agent is discouraged from venturing to the left” I’m not sure what it means to have “imprecise guidance reward’ that nevertheless discourage the policy from taking bad actions.

Relation to Prior Work: While the related works section is good, I recommend that the authors also discuss the relationship of their work with eligibility traces [1, 2]. In particular, on interpretation of this work is that it is implementing a replacing trace [3] with lambda = 1 and a discount factor that is 1 except at the terminal state. [1] Klopf, A. Harry. Brain function and adaptive systems: a heterostatic theory. No. 133. Air Force Cambridge Research Laboratories, Air Force Systems Command, United States Air Force, 1972. [2] Sutton, Richard S., and Andrew G. Barto. Reinforcement learning: An introduction. MIT press, 2018. [3] Singh, Satinder P., and Richard S. Sutton. "Reinforcement learning with replacing eligibility traces." Machine learning 22.1-3 (1996): 123-158.

Reproducibility: Yes

Additional Feedback: Given the lack of theoretical analysis, it would be useful for the authors to discuss some failure modes of the method. Are there situations in which this method would perform poorly? If so, demonstrating such results empirically. -- after the rebuttal: my concerns were addressed, and so I'm increasing my score to an accept. Regarding the reward regression: my suggest was to add the missing red curves in Figure 3. As a suggestion, trimming down algorithm 1/2 pseudo could allow the authors more space for analysis/discussion brought up in the rebuttal.


Review 4

Summary and Contributions: The authors propose a novel approach for temporal credit assignment problem. They estimate the value of a state-action pair by estimating and averaging the returns from trajectories (induced by a behavior policy) that involves the pair. The estimation can be done by building a lookup table in tabular settings, or Monte-Carlo sampling in high-dimensional settings.

Strengths: The core idea in this paper is simple and straightforward: the value of a state-action pair should be the average returns of the trajectories that have the pair in the path. The proposed algorithm, termed Iterative Relative Credit Refinement (IRCR), is therefore simple to implement (as can be seen in the supplementary code) and integrate into existing off-policy RL methods (TD3, SAC). The experimental results show decent performance in a toy grid-world, standard continuous control benchmark, and a multi-agent environment. Overall, the paper provides a simple yet effective method for temporal credit assignment problem, which is a crucial problem in the RL community. I believe that IRCR can be a simple and strong baseline for researchers and RL practitioners working on this problem.

Weaknesses: While the experimental results are promising in the episodic Mujoco tasks, it is unclear whether IRCR works in a more complicated setting (e.g., hard exploration games in Atari, sparse reward tasks, robotic arm pick-and-place tasks, etc). Specifically, in the Mujoco tasks considered in the paper, the episodic reward can be obtained by performing simple, repetitive patterns (i.e., keep going forward). But often, the solution to an environment may involve action sequences that are hardly overlapped or repetitive (e.g., a robotic agent that needs to pick up a block and place it in a specified location, a maze-exploring agent that needs to pick up a key to open the exit). In such cases, the algorithm needs to correctly associate the non-repetitive events with the final return to solve the problem. I wonder if IRCR works in such cases as well. It would be nice if the authors can answer the following questions, which can allow people to understand and assess the proposed approach better: - I am impressed with how IRCR can work so well even with Monte-Carlo sampling, especially when combining it with SAC and TD3 in which replay buffers are involved. My intuition is that, as the replay buffer contains trajectory returns collected by outdated policies, the MC estimates of the guidance reward should be pretty unstable. So how did it work so well in practice? - Figure 3: why use only SAC? How about the results for TD3 and TD3 (IRCR) - Section 4.2: Why use a multi-agent setting? From my understanding, the formula is for temporal credit assignment. The motivation for testing it in a multi-agent setting is not clear to me. Are you trying to show that IRCR is also applicable for multi-agent credit assignment problem? What is the modification required to fit IRCR into this setting? - How does IRCR work in hard exploration games in Atari (defined in [1])? [1] “Unifying Count-Based Exploration and Intrinsic Motivation (NeurIPS 2016)”, Bellemare et al.

Correctness: The proposed approach and claims seem correct to me, and the empirical results seem to support their claims. The experiment results are also obtained from using multiple (5) seeds so the results look convincing to me.

Clarity: The paper is well-written and easy to follow.

Relation to Prior Work: Related works are properly discussed and presented.

Reproducibility: Yes

Additional Feedback: It would be nice if the authors can answer the questions raised above. The overall score may change depending on the answers.

[Author Response · NeurIPS 2020]

We thank all the reviewers for their constructive feedback on improving the paper. We first summarize the new experiments done during the rebuttal period. Due to the limited time/compute (and the need to run multiple seeds), we were able to complete all runs for two MuJoCo environments (*Hopper*, *Half-Cheetah*). We run for 1M timesteps and report the final policy performance (mean $\pm$ std) over the seeds. Due to space constraints, we omit the learning curves but would add them to the revision, along with data for all the other tasks. To put the numbers below in context, SAC (IRCR) achieves $8275_{\pm1279}$ (for Half-Cheetah) and $3214_{\pm279}$ (for Hopper) in the episodic reward setting (Figure 3).

| | EXP. 1 | | EXP. 2 | EXP. 3 | EXP. 4 | | | EXP. 5 | | |
| | SAC (env. rewards) $n$-step returns $(n=10)$ | SAC (env. rewards) $\lambda$-returns $(\lambda=0.9)$ | Distributional-SAC (env. rewards) | SAC (env. rewards) State-augmentation | TD3 (env. rewards) | TD3 (IRCR) | | SAC (env. rewards) | SAC (IRCR) | Random Policy |
|---|---|---|---|---|---|---|---|---|---|---|
| *Hopper* | $626_{\pm281}$ | $566_{\pm101}$ | $292_{\pm31}$ | $358_{\pm123}$ | $235_{\pm37}$ | $2981_{\pm114}$ | *Robotic Arm* | $104_{\pm4}$ | $160_{\pm7}$ | $90_{\pm11}$ |
| *Half-Cheetah* | $-42_{\pm29}$ | $-112_{\pm206}$ | $-24_{\pm16}$ | $-82_{\pm36}$ | $-123_{\pm50}$ | $6844_{\pm918}$ | | | | |

**[R1.]** *Q. Are exploration and credit assignment (due to delayed rewards) the same?* Exploration and credit assignment are two distinct fundamental problems in RL. The former deals with the *discovery* of new useful information, the latter is about efficiently incorporating this information for learning a robust policy. In hard exploration problems, an agent typically obtains zero rewards in each episode unless an exploration impetus is given, whereas in our setting, a reward signal is readily provided to the agent at the end of *every* episode. The focus, therefore, is to train effectively from this delayed feedback and improve upon the credit assignment. We agree that it's important to clarify this distinction and would expand on this further in the revision. *Q. n-step returns and mixed Monte-Carlo?* We include these baselines now; please see EXP. 1 for the results. SAC ($\lambda$-returns) augments the Q-function training with TD($\lambda$), which interpolates nicely between 1-step TD and MC returns based on the value of $\lambda$. We tested with $n=\{3,10,20\}$ and $\lambda=\{0.5,0.9,0.95,1.0\}$. Our conjecture for their low scores with episodic rewards is that the reward delay exacerbates the variance is MC, and bias in TD, to such an extent that using them separately, or mixing them via interpolation, is not sufficient to alleviate the problems in value estimation for credit assignment. IRCR takes a different approach of learning guidance rewards for each time-step, and integrates well with 1-step TD (due to the bias reduction afforded by the guidance rewards).

**[R2.]** *Q. Limitations of IRCR?* Since the reward that the agent optimizes for is different from the original task reward, and is coupled to a behavioral policy $\beta$, for a given MDP, it might be possible to design an adversarial $\beta$ such that optimizing for the resultant guidance rewards leads to unintended behaviors (as per the task rewards). Although not visible in our empirical evaluation, a limitation of IRCR is that a careful adaptation of $\beta$ could be crucial in some domains to avoid this. For instance, if $\beta$ gets stuck in some region of the state-action-space, the learning agent may also get trapped in a local optimum due to *deceptive* guidance rewards. Combining IRCR with methods that explicitly incentivize exploration is a promising approach. We'll include this in the revision. *Q. Unintended output in provided example?* For the given scenario, using IRCR should not make the agent learn $a \rightarrow b \rightarrow e$. Since $a \rightarrow b \rightarrow c$ and $a \rightarrow d \rightarrow e$ are the high return trajectories, the guidance reward for the $b \rightarrow c$ transition should be higher relative to $b \rightarrow e$, making it preferable. *Q. Selection of Baselines?* Some of the methods in Related-works (EBU, HCA, NEC) are proposed for tasks with a *discrete* action-space as they utilize networks that scale with the number of actions. Our Reward-Regression baseline tackles credit assignment in a manner quite similar to RUDDER (as described in 4.1); the available RUDDER code does not handle continuous actions. Self-Imitation is a recent concept introduced to better handle tasks with delayed rewards. *Q. IRCR if there are indeed dense rewards?* In this case, we observe that IRCR reaches similar asymptotic performance compared to the baseline learned with dense rewards but its sample-efficiency is worse; this is because the guidance rewards become more representative gradually, as more data is collected in the MDP.

**[R3.]** *Q. Separate partial observability from reward delay?* We have added a SAC baseline where the state-space is augmented, $\tilde{s}_t = \{s_t, t, \text{done}, \sum_{\leq t} \gamma^k r_k\}$. This restores the Markovian property of the reward function. The low score of this baseline (EXP. 3) points to delayed rewards being the central issue. *Q. C51 baseline for MuJoCo; Reward-regression (RR) on all tasks?* We have added a distributional variant of SAC (EXP. 2). For RR, unfortunately, the code is not open-source, and the authors were unwilling to run additional experiments for us. *Q. Discuss failure modes/limitations?* Please see response to **[R2.]** *Q. Comparison with Eligibility traces?* Thanks for the references and the nice interpretation *w.r.t.* a replacing trace. We'll discuss this in the revision. The baseline in EXP. 1 ($\lambda$-returns) is also relevant here due to the equivalence between TD($\lambda$) (forward-view) and Eligibility traces (backward-view).

**[R4.]** *Q. More domains?* We have added a robotic arm task with episodic rewards: peg-insertion with a 7 DoF Sawyer robot (EXP. 5). We were unable to include further tasks due to limited time/compute and the other rebuttal experiments. *Q. TD3 and TD3 (IRCR)?* We have added these (EXP. 4). *Q. Why multi-agent?* Credit assignment is challenging in the Rover domain because if the agents are outside the observation radius of all POIs – which is typically many timesteps in an episode – then no reward is achieved. Just like single-agent RL, this hampers learning when TD-based MARL is employed. The results on this task show that our approach is generally applicable (*i.e.* not specific to single-agent tasks). No changes were required for the mechanism of calculating the guidance rewards; the architectural modifications are included in Appendix A.2. *Q. Why IRCR works well with data from outdated policies in replay buffer?* Our intuition here is that the return normalization (Line 18, Algo. 2) helps to stabilize training. Specifically, if the trajectory returns from the outdated policies are low compared to the best return seen thus far (this happens gradually over time), the magnitude of guidance rewards for those state-action pairs is low. Hence the agent refrains from visiting those regions.

[Meta-Review · NeurIPS 2020]

This paper tackles the important problem of learning from sparse or delayed rewards. Reviewers liked the simplicity of the proposed approach and found the results to be impressive. The rebuttal addressed most of the main concerns and included important missing baselines and the reviewers reached a consensus that the paper should be accepted. The authors are encouraged to improve the final manuscript based on the reviews. Results on widely studied sparse reward tasks would have made the paper even stronger but the current evaluation warrants acceptance as a poster.